# "Our Freedom in Christ": Revisiting Pauline Imagery of Freedom and Slavery in His Letter to the Galatians in Context

## Albert L. A. Hogeterp

Department of Old and New Testament Studies, University of the Free State, Bloemfontein 9300, South Africa; alahogeterp@gmail.com

**Abstract:** The Letter to the Galatians is a polemical correspondence about the course of gospel mission that is at stake in the view of the apostle Paul. When Paul represents his own contacts with the Jerusalem church, he defends "our freedom which we have in Christ Jesus" (Gal 2:4). In his aim to focus on the unity of all in Christ, Paul even goes at lengths to state that there is no difference between slave and free (Gal 3:28), while polemically associating both a former state of unbelievers (Gal 4:8) and the emphasis by missionary opponents on circumcision and the covenant of the law (Gal 4:12–31) with bondage and slavery. Yet, what did freedom (ἐλευθερία, Gal 2:4) and its opposite, slavery (δουλεία, Gal 4:24, 5:1), exactly mean in the ancient world in which Paul and his readers lived and communicated? Jews, Greeks, and Romans did not necessarily mean the same by these terms, nor did freedom necessarily mean exactly the same as modern conceptions of the term. This paper aims to contextualize Paul's imagery with a view to biblical traditions, early Jewish notions of freedom, and Graeco-Roman registers of discourse, taking into account historical, literary, linguistic, and rhetorical-critical contexts of interpretation and revisiting the language of freedom and slavery with a view to insights from linguistic anthropology. The paper then revisits the Pauline position of "freedom in Christ" in relation to previous hypotheses of Paul's gospel mission.

**Keywords:** freedom; slavery; Pauline discourse; Galatians; contextual biblical interpretation

## 1. Introduction

In his Letter to the Galatians, Paul the apostle aims to uphold the gospel that he had previously preached to his Galatian audience, while dissuading them from believing a different gospel brought to their attention by missionary opponents (Gal 1:6–9). In the course of his argument to persuade his readers, Paul comes to talk about his Jewish background and past life as a persecutor of the church (Gal 1:13–14), his call as an apostle, and his early missionary years, including a delicate search for reconciliation with the Jerusalem church (Gal 1:15–24). Paul goes at length to describe his missionary agreement with the pillars of the Jerusalem church regarding his own gospel mission to the Gentiles (Gal 2:1–10). This description also sets the stage for confronting inconsistencies of opponents to Paul's mission (Gal 2:11–14.15–21). Paul describes them as "false brothers" in the following way in the context of his second visit to Jerusalem around 51 CE:

> But because of false brethren secretly brought in, who slipped in to spy out our freedom which we have in Christ Jesus, that they might bring us into bondage— to them we did not yield submission even for a moment, that the truth of the gospel might be preserved for you (Gal 2:4–5, RSV).

When Paul refers to "our freedom in Christ", ἡ ἐλευθερία ἡμῶν ἣν ἔχομεν ἐν Χριστῷ Ἰησοῦ, it is in severe tension with the actions of his opponents, whose described intent is not only the act of "subjecting", δουλόω, but its superlative form, καταδουλόω, which may denote the act of "subduing in spirit or mind" (Montanari 2015, p. 1051).

What would freedom and slavery mean to the Galatian readers? How may the "freedom in Christ" be interpreted in the contexts of daily life of Paul's readers in the early

Roman era? And what idea of "slavery" does Paul's discourse of freedom exactly oppose? This paper aims to examine Paul's antonyms "freedom", ἐλευθερία, and "slavery", δουλεία, in context in order to revisit earlier hypotheses of an antagonism between a "law-abiding" gospel mission of Paul's opponents and Paul's supposedly "law-free" gospel mission, which also concerns the interpretation of passages in Galatians (Gal 2:19, 3:10–14.23–29, 5:1).[1]

Freedom and slavery would not necessarily mean the same for Jews, early Christians Greeks, or Romans. For instance, in Jewish circles of political influence and leadership, "freedom" and "slavery" could mean something radically different for people of the so-called Fourth Philosophy, who opposed Roman encroachment on Judaea as early as the census of Quirinus by 6 CE (Josephus, *Ant.* 18.1–4), than it did for the leading figures of the Judaean ruling classes, whose notions of freedom would have been more or less in line with a status quo of law and order (Josephus, *J.W.* 4.319, 4.358; cf. Section 3.2 below). "Freedom" could mean something else in the speech of Roman senator Sentius Saturninus (Josephus, *Ant.* 19.167–184) at the time of Claudius' ascent to power as emperor than it would mean for those who acclaimed Claudius as emperor (see Section 4.1 below). "Freedom" and "slavery" could again have different shades of meaning for Greek people under Roman imperial rule, whose sense of identity could employ cultural strategies to adapt to conflicting perspectives.[2]

Another question concerns the issue of how Paul's "freedom in Christ" may be situated on a diverse spectrum of political, philosophical, and religious ideas of freedom and slavery in the early Roman era and how this may add to further interpretation of the apostle's Letter to the Galatians. At first hand, freedom of speech, παρρησία, appears to be an important ingredient to the notion of freedom in Paul's Letters (2 Cor 3:12, 7:4; Phil 1:20; Phlm 8), as well as in the Graeco-Roman world (Keener 2019, p. 114). Yet, in Galatians, freedom of action in the mission seems to be more at the forefront of Paul's description of his initial agreement with the leaders of the Jerusalem church (Gal 2:1–10 at 2:4). Its further interpretation depends on the contextualization among registers of discourse in which the term "freedom" and its antonym "slavery" is used, ranging between political, philosophical, historical, religious, and colloquial settings of communication. To be sure, there is no neat disjunction between political and philosophical/religious ideas about "freedom" and "slavery", since Paul's gospel of Christ also antagonized the "wisdom of this age" along with the "rulers of this age" (1 Cor 2:6–8; RSV).[3] The broader context of society and its vices was so obvious to Paul that to negate it would even mean "you would need to go out of the world" (1 Cor 5:10, RSV). As such, the examination of a broader spectrum of ideas and registers of discourse may help to put Pauline ideas on freedom and slavery into perspective.

This paper aims to highlight a contextual understanding of "freedom" and "slavery" in Paul's Letter to the Galatians where the imagery is very multifaceted. This Pauline imagery ranges from "freedom in Christ" (Gal 2:4, 5:1) to the annulment of distinctions in Christ, including that between slave and free (Gal 3:28); to his notions of the law as a "guardian" for people in states of underaged immaturity and slavery (Gal 3:24–25, 4:1–3) as opposed to divine sonship and heirdom in light of Christ as Son of God (Gal 4:4–7); to the association of idolatry with enslavement (Gal 4:8–9); to the allegory of Hagar and Sarah, referring to the twofold sonship of a slave and of a free woman, respectively (Gal 4:21–31); to freedom in Christ as opposed to a yoke of slavery (Gal 5:1); to the Galatians' call to freedom to be servants to each other in love of the neighbor, thereby fulfilling the whole Law (Gal 5:13–14).

In what follows, this paper will first outline methodological considerations on previous Pauline scholarship and on models of interpretation regarding intercultural communication and linguistic anthropology (Section 2). The discussion of this paper will then turn to biblical and postbiblical contexts of discourse on freedom and slavery (Section 3). This section will explore contexts that may be pertinent to Paul's views, as they relate to his use of Scripture, to the Jewish contexts of his past, being schooled in Pharisaic traditions (Phil

3:5), and of his present as an apostle of the Gentiles learned in Jewish traditions and yet in conflict with rival missionaries and perspectives from the Jerusalem church. The next section will address Graeco-Roman contexts of practice and thought relating to freedom and slavery (Section 4). This paper will then revisit the Pauline imagery of freedom and slavery in his Letter to the Galatians (Section 5) and ultimately turn to an evaluation and conclusions (Section 6).

## 2. Methodological Considerations

### 2.1. Previous Pauline Scholarship on Freedom and Slavery

As a noun, ἐλευθερία, the Greek word for "freedom", occurs in the Pauline Letters (Rom 8:21; 1 Cor 10:29; 2 Cor 3:17; Gal 2:4, 5:1.13) and in the so-called Catholic Letters (Jas 1:25, 2:12; 1 Pet 2:16; 2 Pet 2:19). The antonym of ἐλευθερία, namely δουλεία, denoting "slavery", further occurs only in epistolary texts among the canonical New Testament writings, most of all in the Pauline Letters (Rom 8:15.21; Gal 4:24, 5:1; Heb 2:15). The level of abstraction in discourse about freedom and slavery is thereby most intensely present in Pauline thought.

In a daily life context of talking about one particular slave, Onesimus, Paul knew perfectly well to express the difference between acts "by compulsion", κατὰ ἀνάγκην, and acts "out of one's own free will", κατὰ ἑκούσιον (Phlm 14). Compulsion and its absence as a precondition for freedom also recur in Paul's Letter to the Galatians, as well as his use of the verb ἀναγκάζειν (Gal 2:3.14, 6:12). To be sure, Paul did not advocate the institution of slavery, omnipresent in the Graeco-Roman world. The apostle returned Onesimus to Philemon "no longer as a slave but more than a slave, as a beloved brother" (Phlm 16, RSV). In his Letter to the Galatians, the apostle further envisions the dissolution of distinctions in Christ Jesus, including that between slave and free: "There is neither Jew nor Greek, there is neither slave nor free, there is neither male nor female; for you are all one in Christ Jesus" (Gal 3:28, RSV). Here, Paul voices an idea of the unity of the creation from a Christological perspective (cf. Rom 8:18–25.37–39), which he elsewhere describes as the "glorious freedom of the children of God", ἡ ἐλευθερία τῆς δόξης τῶν τέκνων τοῦ θεοῦ, which sets the creation free from an "enslavement to corruption", ἡ δουλεία τῆς φθορᾶς (Rom 8:21). The call to freedom in Christ that Paul envisages is normatively bound in its good use to "be of good service to one another",[4] δουλεύετε[5] ἀλλήλοις, in love rather than being the occasion for its abuse through the flesh (Gal 5:13).

In order to fine-tune our understanding of Paul's imagery of freedom and slavery in his Letter to the Galatians, which also entails discourses on Jews and Gentiles, it is important to note interpretive positions in Pauline scholarship on Paul's apostleship between Jews and Gentiles. There are various schools of thought with different ideas about this question. For various studies, Paul's Jewish background matters for the interpretation of his Letters in that Paul's Jewish identity also permeates his call as an apostle of the faith in Jesus Christ (see, e.g., Frey 2007, pp. 285–321; Niebuhr 1992; Boyarin 1994; Eisenbaum 2009; Tiwald 2008; Hogeterp 2005, pp. 87–108). Certain studies even go one step further, interpreting Paul within Judaism and considering his discourses as an intra-Jewish matter of perspective (Nanos 2017a, 2017b, 2018; Boccaccini et al. 2016; Tomson 1990). On the other hand, there are also studies that rather consider Paul outside Judaism, studying his Letters as discourses within a Graeco-Roman context, which have more affinity with the socio-literary sphere of Greek popular philosophy rather than with apocalyptic milieus of early Judaism.[6] As such, it is the thesis of this paper that Paul was a man of both worlds: the Jewish world of the Law, the Prophets, and other writings, as well as ancestral traditions, and the Graeco-Roman and Jewish Hellenistic worlds of popular philosophy.[7] These may be bridged by studying his Letters from the perspective of intercultural communication (see further Section 2.2.1 below).

Paul's ideas on freedom and slavery across his Letters have remained relatively understudied. The question regarding what Paul's ideas about "freedom in Christ" would practically mean in contexts of daily life and discourse regarding freedom and slavery in



antiquity has not always been a major concern to Pauline theology.[8] Freedom and slavery have rather been an aside on "practical ethics" than a main issue of discussion in their own right. Major categories of Pauline theology have usually been theology, biblical anthropology, eschatology, soteriology, Christology, pneumatology, ethics, and ecclesiology.[9] Of course, theology should not concern itself with categories, since the abolition of slavery as a societal institution is a fortunate fact since the nineteenth century. Yet, this does not mean that freedom and slavery in Paul's Letters would be a totally disconnected world of thought. Issues of oppression, (self-)enslavement, and manipulation to positions of inferiority may still have to be addressed in political and juridical terms in order to uphold the norm of civic freedom, even in today's world.[10] From a historical perspective of antiquity as the primary context to the Pauline correspondence, the discussion of freedom and slavery may not so much be a matter of the classical theologian's history of Christianity or historical theology as it is a matter of a "people's history of Christianity".[11] The question whether and in which way Paul's discourses on freedom and slavery are contingent on historical realities of freedom and slavery in the Graeco-Roman world has only relatively recently become the focus of historical-critical interpretation.[12]

When considering freedom in Paul's Letter to the Galatians, one should note that various types of freedom could possibly be in view of antiquity and of Pauline thought in particular:

(a) Freedom of belief (cf. Fuchs 1949) Freedom of belief may be exemplified in Galatians with regard to Paul's initial agreement with the Jerusalem church about preaching the faith, which the apostle insists is not based on accommodating men but on his own revelation (Gal 1:10–12, 2:1). Liberty, ἐλευθερία, and good faith, πίστις, are further represented as correlated values by Philo of Alexandria (*That Every Good Person Is Free* 118).

(b) Freedom as free will or freedom of choice (cf. Müller 1926, p. 177) This is also a concept in Hellenistic philosophy, ranging from Epicurean to Sceptic and Stoic thought.[13] In varying ways, the free will also played a role among the Jewish schools, as described by Flavius Josephus, who observed that the Pharisees envisaged an intricate balance between fate (ἡ εἱμαρμένη) and human will (τοῦ ἀνθρωπείου τὸ βουλόμενον) or determinism vs. indeterminism in view of divine providence: "(they postulate that) it was God's good pleasure that there should be a fusion and that the will of man with his virtue and vice should be admitted to the council-chamber of fate" (*Ant.* 18.13; translation from Feldman 1965, pp. 11, 13). Even in an ancient Jewish sectarian context of a voluntary association, some sense of free will is implied by entrance formulas with the Hebrew verb נדב denoting the act of volunteering (1QS 1.7, 5.1), even though further authority rests with the hierarchical community structure of the Dead Sea Scrolls movement in such a context to remain steadfast in all the commandments "in compliance with his (God's) will (לרצונו)" (1QS 5.1; García Martínez and Tigchelaar 2000, p. 79).[14] In Paul's thought, "God's will", τὸ θέλημα τοῦ θεοῦ, is related to human deliverance from evil which, according to the apostle, defines his own age ("the present evil age", Gal 1:4, RSV). For Paul, divine predestination determines his role to be an apostle to the Gentiles by being called through divine grace and by receiving divine revelation of God's Son (Gal 1:15–16). The human free will resides in Paul's response to this personal calling and revelation with eagerness and in "freedom which we have in Christ Jesus" (Gal 2:4, cf. Gal 2:10).

(c) Freedom as economic independence or autonomy[15] In Paul's thought, this type of freedom consists of communal solidarity to "remember the poor" (Gal 2:10).[16] Autonomy or sufficiency (αὐτάρκεια) is also theologically motivated by Paul, with reference to Psalm 112:9 in 2 Corinthians 9:9, which is preceded by the following observation in 2 Corinthians 9:8: "And God is able to provide you with every blessing in abundance, so that you may always have enough (αὐτάρκεια) of everything and may provide in abundance for every good work" (2 Cor 9:8, RSV). Lack of depen-

dence was also a broadly shared notion of freedom in ancient Greek philosophical thought.[17]

(d)　Freedom of speech[18] Freedom of speech or outspokenness, παρρησία, explicitly occurs in some of Paul's Letters (2 Cor 3:12), which further denotes states of confidence (2 Cor 7:4) and openly present courage (Phil 1:20). Paul's writing "with such large letters" (Gal 6:11) is frequently explained by special emphasis on handwriting style and epistolary practices.[19] Yet, this could also be a proverbial example of boldness and outspokenness on the part of Paul as applied to writing. Freedom of speech is that which Paul paradoxically had while in custody in Rome, according to Acts 28:31. Frankness or outspokenness, παρρησία, is also a value correlated with wisdom and righteousness in early Jewish sapiential tradition (LXX Prov 1:20, 10:10; Wis 5:1).[20]

(e)　Freedom of action In Hellenistic and Jewish Hellenistic philosophical discourse,[21] freedom of action is designated as τὸ αὐτεξούσιον (Müller 1926, p. 179), independence, or, more specifically, ἡ αὐτοπραγία, independent action.[22] Freedom of action refers to Paul in relation to his commission as an apostle to the Gentiles in agreement with the Jerusalem church (Gal 2:1–10). Conditional on the affirmation that Paul had been entrusted (πεπίστευμαι, Gal 2:7) the apostleship to the uncircumcised and that he had been empowered with grace by God (χάρις, Gal 2:9), Paul's freedom of action is determined by the absence of compulsion (Gal 2:3), by persistence in his commission without yielding to submission to "false brothers" (Gal 2:4–5), by no additional imposition of anything related to the uncircumcised (Gal 2:6), and by the right hand of fellowship (Gal 2:9).

It depends very much on context, the immediate context of the text, and broader contexts of historical, philosophical, and religious discourses in ancient society, which idea of freedom the apostle Paul would have in mind when he juxtaposes it with its antonym "slavery".

When applied to discourses on justification by faith and works of the law, Paul's ideas on freedom and slavery in Galatians have been divergently interpreted. According to one line of interpretation, the "law of Christ" (Gal 6:2) would ultimately replace the "law of Moses" (Gal 5:14), for which there would be no need in the "new life" in Christ.[23] In this line of thought, "freedom versus slavery, promise and spirit versus flesh" concerns the juxtaposition of freedom in Christ as opposed to enslavement "under law" (Esler 1998, pp. 208–9). According to another interpretation, fulfilment of the "whole law" (Gal 5:14) was common to both sides, but divergently understood by Paul and by his missionary opponents (Dunn 1993, pp. 290–91). A more recent intermediate position tries to balance "fulfilment of a new law, the law of Christ" as a "polemical circumlocution" in Gal 6:2, with the idea that both Gal 5:14 and 6:2 presuppose fulfilment of the law of Moses (Wilson 2007, p. 104).

Contextual biblical interpretation may help to fine-tune our understanding of Pauline discourse on freedom and slavery in Galatians. It may also help to avoid the danger of reified categories, which would equate "faith versus law" with "freedom versus slavery" and thereby produce a problematic simplification of the contexts of thought that may be in view with Paul and his ancient readers. In antiquity, the reality of slavery was not abolished with the Christian faith, as exemplified by injunctions in the deutero-Pauline Letters for slaves to obey their earthly masters (Col 3:22, Eph 6:5–6; cf. Martin 2010, pp. 221, 224). On the other hand, the law of Moses could rather be associated with divine deliverance from a state of slavery in Egypt, with God having "shattered the bond of your yoke (ζυγός) and led you with boldness" (LXX Lev 26:13, NETS), as part of the Holiness Code (Lev 17–26). In this respect, it would not sound logical to associate the law without further ado with "a yoke of slavery" in contrast with "freedom in Christ" in Galatians 5:1, even though a hermetic universe of interpretation with compelling emphasis on works of the law could be the object of Paul's polemic. Thus, ancient realities and perceptions should also alert us against oversimplifications and reified categories of thought and give occasion to rethink what Paul would have in mind under which circumstances.

Before turning to contexts of thought about freedom and slavery, some observations are in place regarding intercultural communication and insights from linguistic anthropology as models of contextual biblical interpretation.

*2.2. Models of Interpretation*

2.2.1. Intercultural Communication

As we noted before (Section 2.1 above), Paul may be considered as a man of two worlds, that of Judaism and his contacts with the Jerusalem church and that of the socio-literary spheres that he shared with his Greek readers. In order to understand Paul's apostleship, as one who aimed to bridge these worlds, our approach should also make room for intercultural comparison. This has been termed a "model of intercultural communication" by P.F. Esler, who took into account social identity approaches to intergroup comparison regarding "Graeco-Roman philosophical and literary influences", "Israelite law, (monotheistic) theology and tradition", and "popular views on religion, philosophy and ethics".[24] Cultural diversity has also been the object of a more recent investigation by K. Ehrensperger, who surveyed various paradigms of cultural encounters, critiquing both the old paradigm of "Hellenism" and a new paradigm of "cultural hybridity", and who proposed to read Paul's Letters in light of bilingualism and biculturalism (Ehrensperger 2013, pp. 17–38). Ehrensperger also warned against emphasizing one dimension ("Jewish, 'Greek or barbarian'") of the intercultural communication at the expense of the other, insisting on the interplay between interlinked dimensions (Ehrensperger 2013, pp. 214–24 at 219).

Interpreting Paul as an apostle at the crossroad of cultures also requires this sensitivity to intercultural communication, which may further involve cross-linguistic comparison of terms across corpora of texts. Indeed, next to his Greek, Paul also employs bilingual Aramaic phrases (μαράνα θά, 1 Cor 16:22; ἀββὰ ὁ πατήρ, Rom 8:15, Gal 4:6; cf. van Unnik 1973, pp. 129–43) and some language use and phrases, such as "works of the law", ἔργα νόμου (Gal 2:16, 3:2.5.10; Rom 3:20.27.28, 4:2.6, 9:12.32, 11:6), intersect with Hebrew in the Dead Sea Scrolls, such as מעשי התורה, "works of the Law" or "precepts of Torah", in 4QMMT C 27.[25] Yet, independently of specific traces of bilingualism, contextual biblical interpretation also depends on cross-linguistic comparisons of concepts and ideas, in our case freedom and slavery.

2.2.2. Insights from Linguistic Anthropology

With regard to the linguistic side of this investigation, our interpretation may further benefit from insights from linguistic anthropology. The comparison of the literature and literary types with regard to their interaction with society has been considered the domain of linguistic anthropology and of the concept of metadiscursivity in recent applied New Testament scholarship (Robertson 2016, pp. 111–18). This metadiscursivity—the study of the intersections between a discourse and socio-literary contexts—is of particular importance for textual comparison where interdiscursivity can be discerned.

Important concepts of linguistic anthropology may help to put intercultural comparisons of Paul's language of freedom and slavery into perspective. One concept is that of the "speech community", which sets the bounds for discursive activities and determines distinctions between insiders (who belong to a speech community) and outsiders (whose sense of identity differs from that of the speech community.[26] For instance, when Paul uses the phrase "works of the law" (Gal 2:16, 3:2.5.10), he indicates that he is fully aware of a speech community where this claims center stage. Yet, at the same time, the apostle distances himself from the idea that "works of the law" are at the heart of justification, thereby putting himself at a distance from such a speech community, even though he does not abrogate the notion of the fulfilment of the law (cf. Gal 5:14). Paul's understanding of law is different from that of an imagined speech community for whom "works of the law" claim center stage, addressing his readers as "spiritual beings", οἱ πνευματικοί, and rephrasing the law elsewhere as being "spiritual" (ὁ νόμος πνευματικός ἐστιν, Rom 7:14).

Another important concept is that of the "registers of language", which can be studied from a metadiscursive perspective as the intersection of repertoires of language with social practices (cf. Agha 2004, pp. 23–45). As applied to Paul, his repertoire of language may vary in sections where he counters the idea of "works of the law" claiming center stage (Gal 2:15–21, 3:1–18) from sections where he explains what it means to be called to freedom (Gal 5:13–15) and to be led by the Spirit (Gal 5:16–24). To be called to freedom does not abrogate the law, rather brings about its fulfilment in Paul's view (Gal 5:14). Laws against trespasses do not address the fruit of the Spirit (Gal 5:22), for "against such there is no law" (Gal 5:23, RSV). Close readings of Galatians should thereby be attentive to "distorting" or "stereotyping" the effects of registers of language.

### 3. Freedom and Slavery in Context: Biblical and Postbiblical Traditions

*3.1. Biblical Traditions*

Scripture and tradition are of primary importance for mapping Paul's position at the crossroads of cultures (cf. Ehrensperger 2013, pp. 140–74). In his Letter to the Galatians, following the introductory chapters 1–2 on relations with Judaism and the Jerusalem church, Paul abundantly cites Scripture, ranging from passages of the Pentateuch[27] to the Prophets (Hab 2:4 in Gal 3:11; Isa 54:1 in Gal 4:27). It may thus be instructive to highlight ideas about freedom and slavery in biblical traditions.

The Pentateuch reflects contexts of ancient Israelite society in which freedom and slavery are part of the narrative and the legal materials regarding bondage, release, and deliverance from slavery. The primary example of deliverance from slavery is the Exodus story (Exod 1:14, 3:9–10.17), where the release from bondage in Egypt is coupled with serving God, who leads Israel out of Egypt. There are also biblical laws concerned with the prohibition to oppress the stranger in one's midst: "You shall not oppress a stranger; you know the heart of a stranger, for you were strangers in the land of Egypt" (Exod 23:9, RSV). Egypt has been described as "the house of bondage" for biblical Israel in Exod 20:2. Exodus 21:2 stipulates the manumission of a Hebrew slave after seven years of service, while Exodus 21:26–27 mandates manumission in cases of physical harm done to a slave, male or female. The Book of Deuteronomy partly reduplicates this (Exod 21:2//Deut 15:12) but also elaborates on material support for a released slave (Deut 15:13–15). Deuteronomy 23:15–16 further adds a law on the humanitarian obligation to let a runaway slave live free from oppression.

Further, the biblical Holiness Code (Lev 17–26), which includes the commandment of love of the neighbor, cited by Paul in Galatians 5:14, also envisages divine providence for God's people in terms of deliverance from slavery in Egypt (Lev 26:13), which, in the Greek Bible translation, is accompanied by shattered bonds of people's yokes and by boldness, παρρησία (of action and speech, LXX Lev 26:13). Paul's citation of legal passages from the Pentateuch in Gal 3:10–14 aims to drive home the idea that any compelling argument of complete reliance on works of the law only ends up with the curse of the law when applied to Jesus Christ. Yet, this does not necessarily preclude or negate that those who are called to freedom in Christ may fulfil the law (Gal 5:13–14). Nor does this line of argument negate the possibility of a spiritual sense of the law (cf. Gal 6:1; Rom 7:14) that may relate to freedom from slavery.

The prophets are among Paul's prooftexts to support his point of justification by faith (Hab 2:4 in Gal 3:11), which also elaborates on the prior exegesis of Genesis 15:6 in Galatians 3:6. With "Jerusalem above (which) is free and she is our mother" in mind (Gal 5:26), Paul cites Isaiah 54:1 in Galatians 5:27. This Isaianic verse is part of a "Song of assurance to Israel" (Isa 54:1–17). According to an extensive study on intertextuality of Galatians with Isaiah by M.S. Harmon, the bottom line of Paul's concluding verses with a wish of peace and mercy on the Israel of God (Gal 6:16) also has an Isaianic background in the same song, i.e., alluding to Isaiah 54:10 (Harmon 2010, pp. 236–38, 265).[28]

*3.2. Freedom and Slavery as Political Values in Early Judaism*

Next to biblical traditions, postbiblical contexts of thought in early Judaism merit further attention in order to situate Pauline discourse on freedom and slavery in intercultural comparison with other Jewish perspectives, which could range from highly political ideas about freedom and slavery to religious and philosophical ideas.

Freedom could be a political value, but its meaning could radically differ depending on whether one belonged to a speech community that aimed to uphold the status quo or to a speech community that was at odds with the status quo of Israel under Roman power and aimed to subvert or overturn it.

### 3.2.1. The Judaean Ruling Classes

Freedom could feature as a rhetorical value of ancient Jewish leadership. Flavius Josephus describes a chief priest, Ananus, as a leading figure, whose death by the hands of the Idumeans would symbolize the beginning of the end (the fall of Jerusalem), characterizing his core values as freedom and democracy (φιλελεύθερός τε ἐκτόπως καὶ δημοκρατίας ἐραστής, *J.W.* 4.319). Josephus further mentions the death of one Gurion from the ruling class; Gurion was full of a democratic and free spirit (δημοκρατικὸς δὲ καὶ φρονήματος ἐλευθερίου μεστός, *J.W.* 4.358). Even if these are idealized pictures of rhetorical praise,[29] they illustrate values that Josephus's reading audience discursively appreciated.

### 3.2.2. Early Jewish Sectarianism

Flavius Josephus describes early Jewish sectarianism at the time of Quirinius' Roman census under emperor Augustus around the turn of the common era (*Ant.* 18.1–23). As it was instituted to tax registered property, Jewish responses to this Roman census were varied, ranging from shock to gradual acceptance to rebellion (*Ant.* 18.3–4). The latter response was attributed to the so-called "Fourth Philosophy". This movement meant theocracy with their passion for "freedom", τὸ ἐλεύθερον (*Ant.* 18.23), associating subjection to Roman taxation instead with "downright slavery" (ἄντικρυς δουλεία, *Ant.* 18.4). This was a radicalized milieu that reportedly did not shrink from bloodshed for its cause of freedom (*Ant.* 18.5).

### 3.2.3. Contexts of War

From the Maccabean era onward, discourses on freedom and slavery could also be part of struggles against oppression of Israel by foreign powers involving open warfare. A poetic fragment in 1 Maccabees 2:7–13 laments the transition from freedom to slavery (1 Macc 2:11), while 1 Macc 14:26 praises the Maccabean war of Simon and his sons against Israel's enemies, establishing freedom, ἐλευθερία, for it. The Maccabean literature also includes narrations about false promises of the foreign ruler Antiochus Epiphanes that Jerusalem and its holy places would be free (1 Macc 15:7, 2 Macc 9:14). As part of a letter to Aristobulus (2 Macc 1:10–2:18), 2 Macc 2:27–29 envisions Israel's liberation from slavery under the Gentiles.

The aforementioned passion for freedom of the "Fourth Philosophy" concerns seeds of rebellion as antecedents to the Jewish war against Rome from 66 to 70 CE. Josephus describes its strides in popular appeal for civil strife toward independence up to the overturning of the body politic by the time of the Jewish war against Rome (*Ant.* 18.5–11).

Even beyond the first century CE, the political sense of freedom from foreign dominion would be at the forefront of upheaval at the time of the Bar Kokhba revolt. Documentary texts of this age contain formulas such as ישראל לחרת, "to the freedom of Israel", and ירושלם לחרת, "to the freedom of Jerusalem" (cf. Eshel 2003, pp. 93–105).

*3.3. Freedom and Slavery from Jewish Religious and Philosophical Perspectives*
3.3.1. Jewish Hellenistic Perspectives

Writing by the turn of the common era, the Jewish Hellenistic author Philo of Alexandria devotes an entire treatise to the subject *That Every Good Person is Free*, comparing

the religious freedom of following God with philosophical religion (*Good Person* 19–20)[30] and with Stoic philosophy of freedom of the good person from the passions (*Good Person* 21–22).[31] Philo goes at length to put the pursuit of virtue as true freedom in a universal comparative perspective, referring to Greek sages, Persian Magi, Essenes, an Indian gymnosophist, Zeno the founder of the early Stoa, and various poets and prose writers (*Good Person* 73–87, 93–103). In *Good Person* 58–61, Philo repeatedly stresses the lack of compulsion and constraint (μήτ᾽ ἀναγκάσαι μήτε κωλῦσαι, *Good Person* 60), which characterizes the good person who performs virtuous deeds out of free will. The opposite, being led by desire, pleasure, fear, or grief, enslaves the soul to "a host of masters" (μυρίων δεσποτῶν) according to *Good Person* 159 (translation from Colson 1941, p. 101). The lack of coercion as a precondition for freedom also resonates with Paul's emphasis on lack of compulsion (Gal 2:3.6, 6:12). As applied to the Galatian correspondence, Paul aims to dissuade the Galatians from a mixture of desire (θέλοντες, Gal 4:21; ἐπιθυμίαι, Gal 5:16.24) and fear brought about by compulsion (Gal 6:12; φοβούμενος, Gal 2:12).

Freedom is further ubiquitous in Josephus' retelling of the biblical narrative. In his *Biblical Antiquities* (*Ant.* 1–11), Josephus refers to freedom, ἐλευθερία, also in the concomitant sense of deliverance or salvation (σωτηρία). Josephus employs this concept, ἐλευθερία, many more times than the Greek Bible, thereby retelling the biblical stories of the Exodus, the wanderings of Israel in the wilderness, and subsequent events in the land of Israel.[32]

### 3.3.2. Jewish Perspectives from the Scrolls and the Early Rabbinic Literature

Next to Jewish Hellenistic texts, postbiblical Semitic corpora of texts merit brief comparative attention regarding religious and philosophical perspectives on freedom and slavery: the Dead Sea Scrolls, from the late Second Temple period, and the early rabbinic literature that dates from late antiquity but includes earlier Pharisaic-rabbinic traditions.

### Dead Sea Scrolls

The Dead Sea Scrolls do not literally provide reflections on a "Jerusalem above which is free, and she is our mother", as Paul does in Galatians 4:26 (RSV). Yet, the Scrolls do contain a multifaceted array of perspectives on the glory and salvation of Jerusalem as the holy city, ranging from an Aramaic prayer of Tobit (4QpapTob[a] frg. 17 col. 2 and frg. 18) to an "Apostrophe to Zion" (11QPs[a] col. 22) to a "Grace after Meals" (with consolation by one's mother likened to consolation in Jerusalem (4Q434a frgs. 1+2 l. 6)) to peace and blessing for God's people and Zion being part of the language of festival prayer (4QDibHam[a] frgs. 1–2 col. 4 ll. 11–13). Since salvation coincides with freedom in religious perspectives, as noted above regarding Josephus, this is the closest analogy one can obtain in the Scrolls for a notion of a free Jerusalem "above", that is, in its vertical dimension of covenantal relations to God.

### Early Rabbinic Traditions

Early rabbinic traditions may further illuminate Pharisaic-rabbinic conceptualizations of freedom and slavery and religious self-perception. In his recent article with a comparative view of "Law" and the alleged "freedom from the Law" in Romans and Galatians, G. Baltes highlighted examples of contrasting images of the "yoke of the Torah", עול התורה, and other human yokes of subjection to the rule of others, i.e., the yoke of slavery, in the rabbinic literature (Baltes 2016, pp. 265–314 at 294, 301–2). This evidence indicates that the "yoke of the Torah" was never equated with the "yoke of slavery" but rather contrasted to it in Pharisaic-rabbinic religious self-perception. When Paul addresses the Galatians not to submit again to "a yoke of slavery" (Gal 5:1), the "yoke" ultimately concerns the compulsive element (Gal 6:12), the compulsion of "the Gentiles to live like Jews" (Gal 2:14, RSV), in which Paul reproves his opponents' insincerity (Gal 2:13, 4:17, 6:13). For Paul (the former Pharisee (Phil 3:5)), it also would not stand to reason to identify the Law in general with a "yoke of slavery". The apostle rather cites from the Law (Lev 19:18) to illustrate its

fulfilment in love of the neighbor, when the Galatians serve each other of their own accord as they have been called to freedom (Gal 5:13–14).

The status of dependence in relation to age (minor vs. mature) is further correlated with discourses of freedom and slavery in both Paul and the rabbinic literature. In a Mishnaic treatise, "children that are minors" are mentioned in apposition to "slaves whether minors or of age" (*m. Ter.* 7.3; Danby 1933, p. 60). Paul's proverbial language addressing the Galatians further stipulates the intersection between dependence and being underaged: "the heir, as long as he is a child, is no better than a slave, though he is the owner of all the estate" (Gal 4:1, RSV). Paul's Greek word for "child" here, νήπιος, also denotes "minor, underaged" (Danker et al. 2000, p. 671),[33] which serves to highlight the aspect of immaturity (rather than innocence) in contrast with the goal of maturity in faith.

## 4. Freedom and Slavery in Context: Greek and Roman Perspectives

### 4.1. Freedom and Slavery in Civic Contexts and Political Discourse

Freedom and slavery were omnipresent political realities in the Graeco-Roman world. It is therefore not surprising to encounter them in civic contexts across various registers of literary and documentary uses of Greek. In the classical Greek literature, freedom could be the object of political reflection regarding forms of government. In his comparison of democracy and oligarchy, Aristotle refers to one line of thought about democracy in which freedom, ἐλευθερία, is mainly found in democracy together with the value of equality, ἰσότης (*Politica* 1291b). Writing in the late Hellenistic period, Polybius considers the political system of Rome to be superior to that of Carthage with regard to the power base of freedom, ἐλευθερία, which depended on a mercenary force in the case of Carthage but on "their own courage (ἀρεταί) and on the assistance of their allies" in the case of Rome (*Histories* 6.52.5). In Hellenistic Greek inscriptions, freedom, ἐλευθερία, could concern the collectivity of the people (δημός), with a setting in piety (εὐσέβεια) toward the gods and respect (φιλοτιμία) toward other citizens (*Demos Rhamnountos* II 22 (229 BCE) (= SEG 15.111) ll. 1–6) and could be related to democracy, δημοκρατία (*IG II² 559* (303/2 BCE), ll. 13–14; *SEG 36.164* (304/3 BCE), ll. 14–15) or to a city-state's autonomy, αὐτονομία.[34]

At the level of the individual rather than a collectivity, Greek inscriptions also illustrate that freedom plays a highly documented role in the manumission of slaves: "for freedom", ἐπ᾽ ἐλευθερίαι (cf. McLean 2011, pp. 292–97 at 292–93). In Greek papyri, freedom and slavery may also be a figure to illustrate a given message. For instance, a private letter from Alexandria, dated 14/13 BCE, includes the following line: "you know full well that, like (ὡς) a slave wishes to please for the sake of his freedom/manumission (ἐπ᾽ ἐλευθερίᾳ), thus (οὕτω) I also wish for your affection" (*BGU 4.1141*, ll. 23–25; translation my own).

As for the early Roman era of the mid-first century CE, one further example of political discourse may help to put the meaning of freedom in perspective. Flavius Josephus writes about the time of Claudius' ascent to power as emperor, whose rule (41–54 CE) approximately spans the era of Paul's early missionary journeys and part of his correspondence.[35] In dealing with the political deliberations about freedom at this time, Josephus represents a speech of one Sentius Saturninus in the Roman senate (*Ant.* 19.167–184). Josephus voices Saturninus's perspective on the dignity of freedom, ἀξίωσις (*Ant.* 19.167), associating it with "independence of thought", αὐτοτελὴς τῆς διανοίας (*Ant.* 19.168), and with "a country with an independent jurisdiction regulating itself with laws", ἐν αὐτοδίκῳ τῇ πατρίδι καὶ μετὰ νόμων … διαιτωμένη βιωθεῖσα (*Ant.* 19.168). This speech further ascribes a "path of liberty for humankind", τῷ ἀνθρωπείῳ τὸ ἐλεύθερον, to virtue (*Ant.* 19.171). Saturninus's speech describes Julius Caesar in negative terms regarding the destruction of democracy and violence to law and order, characterizing his actions as "setting himself above justice but really a slave to what would bring him private gratification" (*Ant.* 19.173). From this perspective, the imperial age of the pax romana was a compromised state of succumbing to "the seduction of peace and (having) learned to live like conquered prisoners" (*Ant.* 19.181).[36] Even though this speech was readily approved by senators and equites (*Ant.* 19.185), Josephus mentions several factors weighing against this senatorial plea to

remain free from imperial rule: the avidness for gain among the powerful members of the senate; the errors in the past committed by a senate in power; the divergence between the will of the people and the will of the senators; the impracticability of senatorial adminis­tration; the stakes of the military aligning itself with imperial rule; the horrors of collective memory regarding civil strife in Pompey's day (*Ant.* 19.162–163, 224–225, 227–228).

The above example of political deliberations may illustrate how complex the broader picture of talking about freedom in the early Roman era really was. These broader political contexts of the Graeco-Roman world provide frames of reference from which a political sense of freedom could be understood by Paul's Greek readers.

### 4.2. *Freedom and Slavery in Philosophical Perspectives*

Next to civic contexts of political discourse, Graeco-Roman contexts of popular phi­losophy merit some comparative attention regarding freedom and slavery in Paul's Letter to the Galatians.[37] The texts of two popular philosophers in particular have been deemed important comparanda for Paul's Letters: Philodemus of Gadara (ca. 110–40 BCE), who was conversant with Epicurean philosophy, and Epictetus (ca. 50–135 CE), who was a Stoic philosopher.[38] Both Philodemus and Epictetus have recently been considered comparable to the socio-literary sphere of ethical–philosophical writing in which Paul engaged with his Letters.[39] In what follows, we will consecutively turn to comparing Paul with Philode­mus and with Epictetus, not in regard to general socio-literary spheres of rhetorical style but with a view to the specific subject under investigation: freedom and slavery.

### 4.2.1. Freedom and Slavery: Paul and Philodemus

When speaking of the role of the Law before justification by faith in Christ came, Paul observes that the Law was added because of transgressions (Gal 3:19), against which the apostle envisages the consecutive roles of the Law as a guardian, παιδαγωγός (Gal 3:24.25), a steward, ἐπίτροπος (Gal 4:2), and a manager, οἰκονόμος (Gal 4:2). These con­secutive roles are set regarding an underaged person, νήπιος (Gal 4:1), who, even being the heir, "is no better than a slave", οὐδὲν διαφέρει δούλου (Gal 4:1, RSV). Paul figuratively compares this immature state of being prior to full heirdom, full adoption to sonship, with "being slaves (δεδουλωμένοι) to the elemental spirits of the universe" (Gal 4:3, RSV). In the larger context of the Letter to the Galatians, which signals compulsive pressures (Gal 1:6–9, 2:4.13–14, 6:12), this state of being enslaved to the material reality of the world would be tantamount to being impressionable by all kinds of influences, as well as pressures of a hermetic universe of interpreting and imposing the Law on Gentiles on the part of Paul's missionary opponents.

The respective images of guardian, steward, and manager as roles of overseeing a mi­nor may further reflect a social reality in the early Roman era, which may be comparatively illuminated. In this regard, Philodemus provides comparative evidence. In his treatise *On Property Management*, he also comes to speak about slaves and freemen in the household. Philodemus holds a sceptic view of an ἐπίτροπος, property manager, when it comes to teaching people to become righteous: "But if, further, he thought it possible to teach the property manager the capacity of making people just, then I consider him to be saying things similar to [the beliefs that we entertain in] our dreams" (*On Property Management* col. 7 ll. 21–26; translation from Tsouna 2012, p. 27). This illustrates a view that being under guardianship of a property manager would not be helpful to teach people a virtu­ous or righteous life from a philosopher's perspective; that idea would be naive, in one's dreams, so to speak. This comparison between Paul and Philodemus implies that the roles of guardian, steward, and manager yield a mixed picture at best when it comes to teaching righteousness and justification. The negative contours of these roles may also be traced back to Paul's own prior life of having a particular zeal for Jewish traditions, which led him to persecute the church (Gal 1:13–14). Paul contrasts such negative contours with a state of being called by God through grace (Gal 1:15), being known by God (Gal 4:9), and being justified by faith (Gal 4:24).

4.2.2. Freedom and Slavery: Paul and Epictetus

There is a longstanding tradition of comparing ethical teachings in the New Testament, and in Paul's Letters in particular, with Epictetus' ethical–philosophical writing, also regarding ideas of freedom.[40] More recently, selected passages of the Letter to the Galatians have been compared with Epictetus' *Discourses* regarding the socio-literary sphere of their rhetorical style.[41] Epictetus approximates theological discourse with his philosophical monotheism that presupposes a supreme being.[42] In this regard, Epictetus's philosophy comes close to the notion of a supreme "unknown god" addressed by the Paul of Acts in his speech at the Areopagus (Acts 17:22–31). The writings of Epictetus and Paul may intersect in their dialogical style, presenting a line of thought by alternating between questions and answers.

One section of Epictetus's *Discourses*, which specifically concerns the topic of freedom (*Diss.* 4.1), is of particular comparative importance for our investigation. From the outset, Epictetus defines freedom as a way of life out of free will that is devoid of compulsion, hindrance, and violence (οὔτ' ἀναγκάσαι ἔστιν οὔτε κωλῦσαι οὔτε βιάσασθαι, *Diss.* 4.1.1).[43] This understanding intersects with Paul's concern against compulsion (ἀναγκάζειν, Gal 2:3.14, 6:12), against intrusion by "false brothers" (Gal 2:4), and against a violent sense of unsettling (Gal 5:12). Epictetus also envisages scenarios of manumitted slaves who relapse in slavery much worse than the former state (*Diss.* 4.1.35), which appears analogous to Paul's admonition in Galatians 4:8–10 not to "turn back again to the weak and beggarly elemental spirits, whose slaves you want to be once more" (Gal 4:9, RSV). Epictetus understands freedom to be freedom of will and autonomy (ἡ ἐλευθερία αὐτεξούσιόν τι εἶναι καὶ αὐτόνομον (*Diss.* 4.1.56)), while further calling for an alignment with the "workings of the divinity and his direction" (τὰς ὁρμὰς τοῦ θεοῦ καὶ τὴν διοίκησιν, *Diss.* 4.1.100). This philosophical monotheism parallels the co-existence of God's will and human freedom in Paul's thought.

If, in Epictetus' line of thought, freedom consists of control of one's passions and not to desire what belongs to others, analogously Paul urges his Galatian readers not to desire under constraint what belongs to others (the pressures of missionary opponents to let Gentile believers live like Jews and be circumcised) (Gal 1:6–9, 2:14, 5:2, 6:12).

## 5. Revisiting Freedom and Slavery in Paul's Letter to the Galatians

*5.1. Setting the Stage for Revisiting Freedom and Slavery: Galatians 1:13–14 in Context*

Having surveyed various Jewish, Greek, and Roman contexts of thought about freedom and slavery, it is time to revisit Pauline thought on this subject. In agreement with a recent study by G. Baltes, it appears problematic to understand Paul's discourses on freedom and slavery as concerning "freedom from the Law" (Baltes 2016, pp. 307–8) because it does not cover all that the apostle has to say about the Law. When he addresses the Galatians with words from Leviticus 19:18 to urge them to be of good service to each other in their call to freedom, Paul deems the "whole Law" fulfilled (Gal 5:13–14, RSV).[44] Yet, which freedom does Paul then exactly have in mind that leads him to address the Galatians in such forceful, even polemical, terms?

In my view, a close reading of Galatians 1:13–14 may set the stage for revisiting Pauline thought about freedom and slavery in Galatians. When we read Paul's thought in light of the background that drove Paul in his former life to persecution of the church, then freedom is not "freedom from the Law" in any determined sense but freedom from a specific milieu of thought that would drive the interpretation of Law observance in directions of compulsion. This may be explained with a comparative view of Flavius Josephus.

When Paul writes about his own Jewish background as a former persecutor of the church, having been "extremely zealous for the traditions of the fathers" (Gal 1:13–14), this is not just meant as a piece of autobiographical information; it also serves a rhetorical purpose. Far from referring to Paul's breakaway from Jewish traditions at large, it denotes his admonition against that way of life in Judaism that led him to persecution of the church. I have argued elsewhere that the background of Paul as a former persecutor

must have been related to a radicalized milieu that may best be understood as a milieu described as the intrusive "Fourth Philosophy" by Josephus (*Ant.* 18.1–10.23–25; Hogeterp 2006, pp. 197–235 at 225–29). It is described by Josephus as a school established by Judas the Galilean at the time of Quirinius' Roman census under the rule of emperor Augustus (*Ant.* 18.1–4.26). This school "agrees in all other respects with the opinions of the Pharisees, except that they have a passion for liberty that is almost unconquerable, since they are convinced that God alone is their leader and master" (*Ant.* 18.23; translation from Feldman 1965, p. 21). This passion for a theocratic "freedom" reportedly led them to factions and to bloodshed (φόνος, *Ant.* 18.5) for the cause of theocratic liberty, even among fellow citizens (φόνος πολιτικός, *Ant.* 18.8), kinsmen, and friends (καὶ συγγενῶν τιμωρίας καὶ φίλων, *Ant.* 18.23). According to Josephus, their influence rapidly gained ground when the cruel Roman governorship of Gessius Florus (64–66 CE) provoked a desperate revolt against Roman rule (*Ant.* 18.25). Yet, even before that point in history, their growing popular appeal is described by Josephus: "Since the populace, when they heard their appeals, responded gladly, the plot to strike boldly made serious progress; and so these men sowed the seed of every kind of misery, which so afflicted the nation that words are inadequate" (*Ant.* 18.6; cf. *Ant.* 18.9; translation from Feldman 1965, p. 7). Josephus's description thereby implies an ongoing appeal of this milieu of thought in Judaea in the first century CE up to the Jewish war against Rome (66–70 CE).

From a retrospective later didactic viewpoint, early rabbinic tradition relates a division among Pharisees between schools of Hillel and Shammai, between lenient and hardline interpretation of the Law, and between gently responding to an obstinately questioning Gentile and chasing that person away with a measuring stick (b. Shabbat 31a). Even if this is a schematized distinction between schools, it may give a sense of milieus of thought. Zeal for the Law and its strict interpretation could be situated within a milieu of thought of hardliners, while zeal for persecution would be a radicalized milieu of thought, even among hardliners. In that respect, the "Fourth Philosophy" would stand out as a particularly sectarian milieu.

The late Second Temple period was also characterized by apocalyptic milieus of thought in Israel. Apocalyptic thought could take various directions of eschatological hope for righteousness, redemption, and salvation, but one direction toward envisioned bloodshed is described in *Jubilees*, an originally Hebrew text[45] (the date of composition is situated in the mid-second century BCE), in the wake of the Maccabean wars (Wintermute 1985, p. 44). In *Jubilees* 23:20, corruption from the covenant at the time of a "future evil generation" is surrounded by envisioned war "in order to return them to 'the way', but they will not be returned until much blood is shed upon the earth by each (group)" (*Jubilees* 23:20; translation from Wintermute 1985, p. 101). If such an apocalyptic line of thought was also entertained in a first-century CE radicalized milieu, such as the "Fourth Philosophy", it would turn the very ideas of "righteousness" and an "evil generation" on their head. Perhaps Paul's polemical statement "You observe days, and months, and seasons, and years!" (Gal 4:10, RSV), following his remarks on enslavement to weak and beggarly elemental spirits (Gal 4:9), has a dark mirror in such apocalyptic thought about war and bloodshed because of forgetfulness of "the commandments and covenant and festivals and months and sabbaths and jubilees and all of the judgments" (*Jub.* 23:19; Wintermute 1985, p. 101).

The above considerations aim to construe against which type of Judaean and Jerusalemite radicalized milieu Paul aimed his polemic in Galatians. In any case, that which emanated from Paul's background as a former persecutor was aiming at persecution "beyond measure", καθ' ὑπερβολήν, and at destruction, ἐπόρθουν, of the church in Paul's own words (Gal 1:13; RSV),[46] and was "breathing murderous threats," ἐμπνέων ἀπειλῆς καὶ φόνου,[47] in the words of Acts 9:1 (RSV). Even though Paul's missionary opponents in Galatians 1:6–9 would not by themselves have belonged to such a radicalized milieu, most scholars agree that the opponents whom Paul envisaged were Judaizers from Jerusalem.[48] With a Jerusalemite setting in mind, Paul's ideas of compulsion would be different from those of his Galatian readers, whom he warns not to be caught unawares. Paul's Letter to the Gala-

tians does also provide glimpses of external pressures of a "they" group, such as "false brothers" (Gal 2:4), and a contemporary scenario, apart from Paul's own past, of compulsory circumcision "in order that they may not be persecuted for the cross of Christ" (Gal 6:12, RSV).

*5.2. Pauline Polemics: Freedom from a Radicalized Milieu of Law Interpretation*

If we want to obtain a fuller picture of what fueled Paul's polemic in Galatians (which the apostle voiced with his discourses on freedom and slavery), it is further important to follow the hints of intrusive influences in Galatians that could point to the encroachment of a radical milieu of thought in Jerusalem.

In Paul's account of relations with the Jerusalem church, the influence of "false brothers" (Gal 2:4) and "certain men from James" (Gal 2:12) is apparent. Yet, when Paul polemicizes against insincerity about relations between Jews and Gentiles, which would "compel the Gentiles to live like Jews" (Gal 2:14, RSV), this may also concern a larger context of milieus in Jerusalem. There are several indications in Galatians to take the encroachment of radical influences of external pressures on Jerusalem-based circles seriously.

(a)    "false brothers", ψευδάδελφοι (Gal 2:4), secretly brought in to spy out the freedom of Paul's gospel mission. The label "false brothers" is much more severe in its antagonism than sometimes supposed.[49] The intrusive way in which they are presented (τοὺς παρεισάκτους ψευδαδέλφους, "false brothers secretly brought in", Gal 2:4, RSV) is not even parallel to the straightforward way in which Paul subsequently writes about "certain men (who) came from James" (Gal 2:12, RSV). This intrusive presentation (παρείσακτος) somehow parallels Josephus' word, ἐπείσακτος, for introducing the "Fourth Philosophy" as an intrusive school. If, as C.S. Keener notes, the false brothers "were ultimately in cahoots with nonbelieving Judeans who pressured Judean believers (cf. 6:12)" (Keener 2019, p. 118), the intrusive element of spying against the freedom in Christ would possibly serve a purpose of external pressures. In any case, the appellation "false brothers", ψευδάδελφοι, is uniquely Pauline in New Testament Greek (2 Cor 11:26, Gal 2:4), being most squarely the opposite of the family language of "brothers", ἀδελφοί, which the apostle otherwise uses to amicably address his Galatian readers (Gal 1:11, 3:15, 4:12.28.31, 5:11.13, 6:1.18).

(b)    "They make much of you, but for no good purpose; for they want to shut you out (ἐκκλεῖσαι), that you may make much of them" (Gal 4:17, RSV) According to Paul, the "they" group described in this verse exerts a power of exclusion from fellowship to pressurize Galatians to make much of them, yet "for no good purpose", denying them their free will in the interest of a type of communal boundary marking, which is not Paul's gospel of Christ (cf. Gal 4:19; RSV).[50] "It is those who want to make a good showing in the flesh that would compel you to be circumcised, and only in order that they may not be persecuted for the cross of Christ. For even those who receive circumcision do not themselves keep the law, but they desire to have you circumcised that they may glory in your flesh" (Gal 6:12–13, RSV).

(c)    The compulsory circumcision, the fear of persecution, and the glorying of a "they" group in the flesh are all elements that combine into a picture of an abnormal radical milieu of thought about the Law and circumcision. Paul here employs a phrase, ἀναγκάζουσιν ὑμᾶς περιτέμνεσθαι (Gal 6:12), which has a parallel in exceptional wartime circumstances in Galilee narrated by Flavius Josephus. Josephus describes his disallowance of a Jewish attempt of forcible circumcision (τούτους περιτέμνεσθαι τῶν Ἰουδαίων ἀναγκαζόντων) of refugee noblemen from Trachonitis "as a condition of residence among them", εἰ θέλουσιν εἶναι παρ' αὐτοῖς (*Life* 113; Thackeray 1926, p. 45). Against this radicalized world of thought with enforced circumcision (Gal 6:12), Paul emphasizes that "neither circumcision counts for anything, nor uncircumcision, but a new creation" (Gal 6:15, RSV). Another element of the radicalized world of thought is persecution, mentioned here in the passive form (διώκωνται) in Gal 6:12 and also in Gal 5:11 (διώκομαι), which has been related to various possi-

ble sources among Jerusalemite attitudes, nationalistic sentiments, and local attitudes (Keener 2019, pp. 566–67).

What Paul's Letter to the Galatians lacks in further explicit detail regarding Jerusalemite circles apart from hints at compulsion, external pressure, and persecution may be compared with external contextual evidence of Flavius Josephus on the Fourth Philosophy, as we have seen above (Section 5.1 above). The fact that milieus of radicalization in Jerusalem would eventually touch the context of Paul's ultimate visit to Jerusalem may be gathered from the secondary evidence of Luke's story of Paul in Acts 21:17–23:22. Even though the evidence of Acts may by no means constitute primary evidence of Paul's life and thought, the author of Acts did have his sources of information about Paul beyond the timeframe of his letters.[51] His narrative does provide an additional perspective on the subject of radicalized milieus in Jerusalem by the late 50s CE.

The Acts of the Apostles detail specific contexts in Jerusalem with which Paul's gospel mission clashed. The chronology of events in Jerusalem in Acts 21:17–23:22 can be approximately dated to the late 50s CE in view of subsequent reference to a succession in Roman governorship from Felix (52–60 CE) to Festus (60–62 CE) after a lapse of "two years" in Acts 24:27 (cf. Fitzmyer 1998, p. 740). This is almost a decade after Paul's second visit to Jerusalem that he mentions in Galatians 2:1–10, which has been dated around 51 CE (cf. Murphy-O'Connor 1996, p. 8). It is during this decade of the 50s CE, during Felix's governorship in Judaea, that Josephus situates the rise of the Sicarii, σικάριοι,[52] a radical group willing to kill people for revolutionary changes in Judaea and, in particular, in Jerusalem (*J.W.* 2.254).[53] As we will see, this radical group literally figures in the Roman apprehension about the tumult surrounding Paul's presence in Jerusalem, as narrated in Acts.

There are three indications of radicalized milieus in Jerusalem in Acts. First, Acts 21:17–26 narrates Paul's encounter in Jerusalem with James (surrounded by elders), who urges Paul to make ritual amends for the impression that he would cause Jews who are among the Gentiles to forsake Moses (Acts 21:17–26). Here, a "they" group exerting external pressure on the Jerusalem church is clearly in view (Acts 21:20–21). Even so, according to Flavius Josephus, writing about events under Albinus' procuratorship of Judaea (62–64 CE), the ultimate fate of James was ominous (being stoned under the Sadducean high priesthood of one Ananus for alleged transgression of the Law) (*Ant.* 20.199–200).[54] The second specific context of a radicalized milieu is mentioned in Acts 21:37–39, when a Roman tribune cross-examines the identity of Paul. In Acts 21:38, the tribune asks Paul, after they have exchanged a few words in Greek, whether he is not "the Egyptian, then, who recently stirred up a revolt and led four thousand men of the Assassins (σικάριοι) out into the wilderness" (RSV). This interrogation, which is answered by Paul by asserting his own identity as a Jew from Tarsus in Cilicia (Acts 21:39), may be compared with various accounts of Josephus about "a false prophet from Egypt" (*J.W.* 2.261–263) or "a certain impostor" (*Ant.* 20.188) in the same context, where Josephus mentions related brigandry by the Sicarii.

A third indication of a radicalized milieu in Jerusalem at the time of Paul's visit by the late 50s CE is provided by the narration in Acts 23:12–15 about a group of more than forty Judaeans who allegedly "made a plot and bound themselves by an oath neither to eat nor drink till they had killed Paul" (Acts 23:12, RSV), which was overheard by "the son of Paul's sister" (Acts 23:16). It was not the Pharisees of the Jewish council who united in opposition against Paul; according to Acts 23:6–10, they were divided because of Paul's own Pharisaic allegiance. Thus, this is another instance of a radicalized milieu in Jerusalem.

Even though the explicit indications of Acts about radicalized milieus in Jerusalem with which Paul's gospel mission clashed postdate his Letter to the Galatians, the radicalization will not have appeared overnight and the hints at the encroachment of radical influences in Galatians affirms the impression that Paul polemicized against such influences from a "they" group of "false brothers" (Gal 2:4) beyond himself and the pillars of the Jerusalem church. As such, dangers of persecution of apostate and foreign elements by a radicalized milieu of Law interpretation were not behind Paul and his readers, as a mat-

ter which would only concern Paul's past as a persecutor of the church (Gal 1:13–14; Phil 3:5–7), but loomed ominously in the context of writing (Gal 4:29 ἐδίωκεν, 5:11διώκομαι, 6:12 ἵνα . . . μὴ διώκωνται).[55]

## 6. Evaluation and Conclusions

This paper has endeavored to explore the diverse contexts of reality and thought about freedom and slavery in Jewish, Greek, and Roman contexts in order to revisit their meaning in Paul's Letter to the Galatians. Across various cultural contexts, freedom has been associated with a free will and a freedom of choice in Jewish, Jewish Hellenistic, Greek, and Roman contexts of thought and philosophical reflection. Yet, political realities of the day could provoke exceedingly divergent ideas about freedom and slavery in intercultural contexts of communication. Where the Judaean ruling classes would include voices of freedom and democracy in upholding law and order, sectarian radicalized milieus, such as the "Fourth Philosophy" described by Josephus, did not perceive freedom, but rather downright slavery, in submitting to Rome. This paper has contended that it is this politically charged atmosphere of zeal for theocratic "freedom" from the early first century CE onwards to outright radicalization by the 50s CE in Judaea and Jerusalem that informs Paul's polemical tone in Galatians, when the compulsion for Gentiles to live like Jews is apparent through external pressures from Jerusalem.

Paul's "freedom in Christ" (Gal 2:4, 5:1) reflects a radical turn away from his former life as a persecutor of the church and from his immature zeal for the Law (Gal 1:13–14) toward redemption from the curse of the Law (Gal 3:13) by the revelation of God's Son to him (Gal 1:15, 4:4–5). The "curse of the Law" (Gal 3:13) would lead Paul into the double trap of having to condone a curse against Christ because of his crucifixion (Deut 27:26, 21:23)[56] and of conforming with a zeal for the Law which led him to persecute the church, while at the same time claiming "blamelessness" as to "righteousness under the Law" (Phil 3:6). It was to this sense of the Law that Paul claims that "I through the law died to the law, that I might live to God" (Gal 2:19, RSV).

As such, Paul does not advocate a law-free gospel (cf. Gal 5:13–14), let alone a gospel that would leave room for lawlessness,[57] but polemicizes against a radicalized milieu that would drive the interpretation of the Law to directions of condemning the "cross of Christ" (Gal 3:13) and justifying persecution of those who would pose a challenge to the zeal for the Law (Gal 6:12). It is a radicalized milieu of persecution that Paul not only situates in the past in light of his own background as a persecutor of the church (Gal 1:13) but also relates to the present when dissuading the Galatians from compelling arguments for circumcision: "and only in order that they may not be persecuted (μόνον ἵνα τῷ σταυρῷ τοῦ Χριστοῦ μὴ διώκωνται) for the cross of Christ" (Gal 6:12, RSV). As we have seen, Jerusalemite contexts were characterized by radicalized milieus by the 50s CE.

Paul was not alone in his polemics against radicalization within Judaism. The literature of Qumran also includes polemic voices against those who betrayed God's covenant, ותשקרו בבריתו, while saying to themselves "Let us fight his battles, for he has redeemed us", נלחמה מלחמותיו כיא גאלנו (4Q471a (*4QPolemical fragment*) frg. 1 ll. 2–3); of an envisioned "evil generation", דרה באיש, in a position of deceit and violence as opposed to a priestly protagonist of atonement (4Q541 frg. 9 col. 1 ll. 2, 6–7); of envisioned times of judgement in a setting of one who "will go, and will be oppressed (ויתעשק), and he will say: «Let me go to the ends of the earth»" (4Q568 l. 1).[58]

Paul's allegory of Hagar and Sarah (Gal 4:21–31), applied to a contrast between "the Jerusalem above", which is free, ἐλευθέρα ἐστιν, and the "present Jerusalem", which is enslaved with her children, δουλεύει γὰρ μετὰ τῶν τέκνων αὐτῆς (Gal 4:25–26), may further concern the antagonism with a radicalized milieu of thought. Perhaps, analogously with Paul's association of the enslaved Jerusalem with Mount Sinai in Arabia (Gal 4:24–25), Mount Sinai was also claimed by radicalized milieus of Law interpretation. Josephus relates circumstances in Judaea from the late 50s CE onwards, when an Egyptian false prophet led followers through the wilderness back to Jerusalem to overthrow Roman rule

(*J.W.* 2.261–262). Jerusalem was considered "trampled" in the early Roman age, as illustrated by the Qumran sectarian Nahum Pesher (4QpNah 3+4 i 3).[59] Yet, the hardboiled violent factionalism of the first century CE, which Josephus describes as "strife between factions and the slaughter of fellow citizens" (*Ant.* 18.8; Feldman 1965, p. 9), did not beget freedom, rather ruin and further subjugation into slavery. Perhaps Paul's "Jerusaelm above" rather concerned the city of God as imagined in prayers in a vertical dimension of people's covenantal relations with God. The interpretation of Gal 4:21–31 in light of the polemics against a radicalized milieu may further find support in Paul's reference to persecution: "But as at that time he who was born according to the flesh persecuted (ἐδίωκεν) him who was born according to the Spirit, so it is now" (Gal 4:29, RSV). The Pauline contrast between flesh, σάρξ, and Spirit, πνεῦμα (cf. Gal 5:16–26), aligning the Galatians with spiritual people, οἱ πνευματικοί (Gal 6:1), may intersect with a Jewish sapiential perspective on the distinction between a "spiritual people", עם רוח, and "the spirit of flesh," רוח בשר, which "does not know the difference between [goo]d and evil" (4Q417 (*4QInstruction^c*) frg. 2 col. 1 ll. 16–18; García Martínez and Tigchelaar 2000, pp. 858–59).

This paper also yields the conclusion that Paul's categories of freedom and slavery ultimately do not concern a contrast between Christian freedom and slavery under a "yoke of the Law" but freedom for Jewish and Gentile followers of his gospel of Jesus Christ to remain in the state in which they were called (cf. 1 Cor 7:24). In this regard, Paul's perspective would not be so much different from Josephus' wartime response to Jews who would compel Trachonite refugees to be circumcised for their residence among them: "I, however, would not allow any compulsion to be put upon them, declaring that every one should worship God in accordance to the dictates of his own conscience and not under constraint, and that these men, having fled to us for refuge, ought not to be made to regret that they had done so" (*Life 113*; Thackeray 1926, p. 45). Paul's bottom line in Galatians is the wish for peace and mercy "upon the Israel of God" (Gal 6:16), thereby insisting on a vertical dimension of the relation to God, while inverting the phrase "God of Israel" that so often occurs in ancient Jewish Hebrew prayer texts.[60]

**Funding:** This research received no external funding.

**Data Availability Statement:** All relevant data are included in this article with endnotes and references.

**Conflicts of Interest:** The author declares no conflict of interest.

## Notes

1. See Niederwimmer (1966, pp. 168–219) on the Pauline concept of freedom, at 192–212 ("Die Freiheit vom Gesetz"); Conzelmann and Lindemann (2004, p. 239) on Gal 3:1–5:12 as being about "Freiheit vom Gesetz"; Schnelle (1996, p. 129) on the religio-historical position of Galatians as Paul's law-free gospel ("gesetzesfreien Heidenmission"); Wolter (2021, pp. 374–77) also considered "freedom from the Law and the 'law of Christ'" ("Die Freiheit vom Gesetz und das »Gesetz Christi«") as a rubric of discussion in a chapter on "justification from faith" (339–410). Cf. Meeks (2001, pp. 17–27) at 18–20 on the influential "Tübingen school" initiated in the 19th century by F.C. Baur, who hypothesized an antithesis between "Jewish Christianity", represented by the circle around James in the Jerusalem church, and "Gentile Christianity", represented by Paul. See more recently the argument by Fredriksen (2015, p. 638) against the "remarkably enduring" "view of Paul's personal rejection of Jewish ancestral custom" and nn. 3–4 with further bibliography on this scholarly tendency.

2. Cf. Goldhill (2001, p. 1) on Plutarch's *Roman Questions* 273 relating to the conflicting perspective of Roman military power vis-à-vis "enslavement and effeminacy of the Greeks" embodied in their cultural institutions of gymnasia and wrestling schools.

3. For a recent study on Pauline discourse in relation to the politics of his days, see Heilig (2022).

4. The translations "freedom in the slavery of love" or "through love be slaves to one another" in Gal 5:13, as represented by Wilson (2007, p. 98), appear less helpful, rendering an alleged paradox. Yet, the apostle may rather have subversion of institutional slavery in mind, in line with the Jesus tradition that urges followers of Jesus Christ to serve one another (διάκονος εἶναι, Mk 10:43; δοῦλος εἶναι, Mk 10:44; διακονεῖν, Mk 10:45) rather than to lord it over each other (Mk 10:42–45).

5. According to Montanari (2015, p. 551), the verb δουλεύειν may have diverse shades of meaning, not being limited to the state of "being a slave, being enslaved/subjugated" but also meaning "to serve", with a dative "to faithfully follow".

6    Cf. Robertson (2016, pp. 1–9), who considers discussions of Paul as "Greek or Jew" and Paul's Letters as "high rhetoric or Jewish apocalyptic" unhelpful "essentialized" understandings, while associating Paul's literary style most of all with the socio-literary comparanda of Philodemus and Epictetus; Malherbe (1989); Thom (2015, pp. 47–74).

7    Cf. Fredriksen (2017) on Paul as a man who bridged two ancient worlds, a Jewish world of apocalyptic hopes of messianic redemption and a pagan world of beliefs in cosmic superhuman forces intervening with the human environment.

8    When discussing "freedom" as a concept, Niederwimmer (1966) differentiated ancient political and philosophical notions of "freedom" (pp. 1–68) from eschatological conceptualizations of "freedom" in the New Testament, highlighting the Jesus tradition (pp. 150–67), Paul (pp. 168–219), and John (pp. 220–34). However, Pauline discourses on freedom and slavery in Galatians also employ language derived from contexts of daily life and possibly intersecting with philosophical discourse and political realities.

9    For instance, Dunn (1998) did not devote a separate paragraph to the subject of "freedom and slavery" in his chapters on "God and Humankind" (ch. 2), "Humankind under Indictment" (ch. 3), "The Gospel of Jesus Christ" (ch. 4), "The Beginning of Salvation" (ch. 5), "The Process of Salvation" (ch. 6), or "The Church" (ch. 7), but discussed "Liberty and love" (658–61) and "Living between two worlds: slavery (1 Cor 7:20–23)" (698–701) in a few pages as part of his chapter 8 on "How Should Believers Live?" (625–712). Cf. Schnelle (2003, pp. 433–700) regarding Pauline thought in the categories mentioned in the main text above, while, at 287–330, his chapter on Galatians focuses on the understanding of the Law, justification, and ethics. Wolter (2021, pp. 339–410) devotes much attention to "justification from faith" ("Die Rechtfertigung aus Glauben"), including discussion of "freedom from the Law and the 'Law of Christ'" ("Die Freiheit vom Gesetz und das »Gesetz Christi«", § 44). Yet, in Galatians, freedom is defined most of all as "freedom in Christ" (Gal 2:4, 5:1) and as a call to freedom toward one's neighbor in love of the neighbor, in which the whole Law is fulfilled (Gal 5:13–14 at v. 14 citing Lev 19:18) rather than literally speaking "freedom from the Law". Paul's discourse rather counters a hermetic interpretive universe in which the Law justifies people's perspectives up to the point of justifying the curse of "everyone who hangs on a tree" (Gal 3:13 citing Deut 21:23/Deut 27:26).

10    The fact that slavery is no longer an institutional reality as part of regular international world order since the 19th century unfortunately does not preclude crimes such as kidnapping, abduction, forced labor, human trafficking, and sexual slavery, as committed by the Islamic State in the mid-2010s against Yazidi women and girls.

11    See in this regard, Martin (2010, p. 221): "a people's history not only involves but requires the investigation of the lived experiences of enslaved believers".

12    See, e.g., the recent case study on the categories of "free" and "slave" in 1 Cor 9:19 as part of 1 Cor 9:19–23 by Bühner (2023, pp. 200–6) on Jewish identity in situations of slavery in the ancient Roman world at the time of Paul's Letters.

13    Cf. Long, 1986, on freedom of will among Epicureanism (56–61), Carneades (101–4), and Stoicism (167–67, 207).

14    For the concept of the "will of God" in the Qumran literature, cf. the Aramaic רעות אל, "the will of God", in 4Q541 9 i 3.

15    Termed "Freiheit als Gehalt" and "Autonomie" by Müller, 1926, pp. 177–78.

16    On this subject, cf. Longenecker (2010). Cf. Hogeterp (2014, pp. 261–75) at 270 on an Aramaic proverb דכור עני in 4Q569 (*4QAramaic Provebs*) frgs. 1–2 l. 8 that parallels the injunction to "remember the poor" in Gal 2:10.

17    Cf. Keener (2019, p. 115, n. 330) with ancient references.

18    Cf. Keener (2019, p. 114, n. 325) on the relation between παρρησία and freedom in Philo and Josephus. See also Cassius Dio, *Roman History* 43.10.5 on Cato's words addressing his son that he had been brought up in freedom with the right of free speech, ἔν τε ἐλευθερίᾳ καὶ ἐν παρρησίᾳ.

19    Cf. (Keener 2019, pp. 559–63) for a survey of interpretations of Gal 6:11.

20    Cf. (Wis 7:22–23) on various qualifications of the spirit of wisdom, including "free-moving", εὐκίνητον (Wis 7:22), and "unhindered", ἀκώλυτον (Wis 7:23), an adjective related to the adverb ἀκωλύτως following the phrase μετὰ πάσης παρρησίας in Acts 28:31.

21    Cf. Long (1986, pp. 56–61) on "freedom of action" in Epicurus and Epicureanism.

22    Müller (1926, p. 180, n. 1) cited Diogenes Laertius, *Lives of Eminent Philosophers* 7.121, with Zeno, the founder of the Early Stoa, on freedom as ἐξουσία αὐτοπραγίας, "the power of independent action", and Philo, *That Every Good Person Is Free* 21–22 with the same noun αὐτοπραγία and τὸ αὐτοκέλευστον, denoting the state of being "self-motivated, unconstrained, spontaneous". Cf. Montanari (2015, p. 343) s.v. αὐτοκέλευστος.

23    Esler (1998, p. 204): "The moral standards or norms of the law, just like the rest of the law, are not 'taken up' into the new life. They have no further purpose for those who believe in Christ".

24    Esler (1998, pp. 9–28) on intercultural communication and intercultural reading of Galatians at 12.

25    See also מעשי בתורה in 1QS 5.21 and מעשיהם בתורה in 4QS^d 2.1. Regarding 4QMMT, comparison with Paul has been the object of investigation since the mid-1990s. See, e.g., Kampen (1996, pp. 129–44). Cf. Hogeterp (2014, pp. 261–75) for a survey of cases of Graeco-Semitic bilingualism in Galatians in light of the Dead Sea Scrolls.

26    Cf. Morgan (2004, pp. 3–22). Muehlmann (2014, pp. 577–98) at 584 also considers an alternative revised concept of "community of practice" that "considers language as one of many social practices in which participants engage". In distant historical contexts, the text-based study of language may yet benefit from the concept of speech community. Ahearn (2011, pp. 99–118) uses the term "communities of language users".

27 LXX Gen 15:6 in Gal 3:6; LXX Gen 12:2 (ἐν σοί)/Gen 18:18 (πάντα τὰ ἔθνη) in Gal 3:8; LXX Deut 27:26 in Gal 3:10; LXX Lev 18:5 in Gal 3:12; LXX Deut 27:26 and 21:23 in Gal 3:13; LXX Gen 13:15, 17:8, 24:7 in Gal 3:16; Gen 16:15, 21:2.9.10 in Gal 4:22–23; LXX Gen 21:10 in Gal 4:30; LXX Lev 19:18 in Gal 5:14.

28 Yet, Harmon (2010, p. 264) concludes that the messianic age envisaged by Paul in his Letter to the Galatians would liberate from "the bondage of the Law", understanding the fulfilment of the Mosaic Law in Gal 5:14 in light of its redefinition as "law of Christ" in Gal 6:2. However, Paul's argument against being "under law" may rather concern a specific interpretive context of opponents, not necessarily the Law at large.

29 In *J.W.* 4.319 and 4.358, Josephus otherwise refers to the aristocratic descent of Ananus of Gurion. Josephus employs varying characterizations of Jewish forms of government from the time of the Persian king Cyrus until the reign of Antiochus Eupator (164/163 BCE), ranging from "a democratic form of government" (*Ant.* 20.234) to "a mixture of aristocracy and oligarchy" (*Ant.* 11.111).

30 In *Good Person* 19, Philo cites a line from Sophocles to illustrate freedom in philosophical monotheism.

31 Freedom from passions concerns the state of being yoked (καταζευγνύειν) by desires, fears, pleasures, or grief (*Good Person* 18); cf. *Good Person* 159.

32 *Ant.* 2.92, 2.252, 2.281, 2.290, 2.327, 2.329, 3.19, 3.44, 3.64, 3.283, 3.300, 4.2, 4.42, 4.187, 5.34, 5.182, 5.194, 5.214, 5.265, 6.19, 6.20, 6.60, 6.98, 7.95, 7.258, 8.38. The Greek Bible only employs the noun ἐλευθερία once in the canonical books, in LXX Lev 19:20.

33 Paul's figurative language in Gal 4:1–7 should not be mistaken as a categorically negative perspective on childhood as a proverbial state, for elsewhere in his Corinthian correspondence the apostle addresses his readers as "my beloved children" (1 Cor 4:14), one Timothy as a "beloved and faithful child in the Lord" (1 Cor 4:17), and with the injunction "do not be children in your thinking; be babes (νηπιάζετε) in evil, but in thinking be mature" (1 Cor 14:20, RSV).

34 *IG II²35* frg.a ll. 12–13 (Attica, 384/383 BCE), *IG VII 2713* (Boiotia, Akraiphia, 67 CE) l. 43; *SEG 32:469* (Boiotia-Koroneia-Pontza, 161 CE) l. 8.

35 Murphy-O'Connor (1996, p. 8), dates Paul's first visit to Jerusalem after his call as apostle (Gal 1:18) to 37 CE, his second visit to Jerusalem (Gal 2:1–10) to 51 CE, and situates Paul's journeys to Syria and Cilicia (Gal 1:21) in the intermediate period after 37 CE.

36 Translations from Feldman (1965, pp. 293, 295, 299).

37 So far, civic ethics with a view to "honor" and "shame" have been a particular focus of investigation regarding Paul in a Graeco-Roman context. See, e.g., Harrison (2015, pp. 75–118).

38 According to the narrative of Acts 17:16–34 at v. 18, Epicurean and Stoic philosophers belonged to the conceptual orbit of the Greek world of discourse in Athens visited by Paul the apostle.

39 Robertson (2016) compares these three in terms of genre and taxonomy of social-historical context (pp. 89–120), of literary criteria regarding rhetorical style (pp. 121–69), and of socio-historical description of their respective education and lives (pp. 170–214).

40 Cf. Bonhöffer (1911); Bultmann (1912, pp. 97–110, 177–91); Schmitz (1923); Müller (1926, p. 180) on Epictetus' idea of freedom as "autonomy", but contrasts Paul's thought about freedom with a philosophical teaching of freedom, since it is based on preaching faith in Jesus Christ (182); cf. Müller (1926, p. 189).

41 Robertson (2016, pp. 121–69) includes some passages from the Galatian correspondence (pp. 130, 133, 135, 140) in his comparative survey regarding the literary use of rhetorical devices, such as "metaphors/analogies" (Gal 5:1–12, 4:21–31; Epictetus, *Discourses* 2.1.7–9, 12), "caustic injunctions" (Gal 1:6–12; Epictetus, *Discourses* 2.1.30–33), "hyperbole" (Gal 4:12–20), and "systematic argument" (Gal 3:13–26; Epictetus, *Discourses* 1.2.5–10).

42 Cf. (Long 1986, p. 235), "Epictetus calls God the father of mankind".

43 At a later stage in this section, in *Diss.* 4.1.128, Epictetus recapitulates this same understanding of freedom.

44 In this respect, even non-Jews appear not entirely "law-free" in Paul's gospel, even though, as Baltes (2016, p. 308) asserts, they are free from the "curse of the law".

45 Hebrew manuscripts of *Jubilees*, 1Q17–18 (1QJub^a–b), 2Q19–20 (2QJub^a–b), 3Q5 (3QJub), 4Q216–224 (4QJub^a–h), and 11Q12 (11QJub), have been preserved at Qumran, of which the palaeographical dates range from the second half of the second century BCE to the mid-first century CE. Cf. 4Q225–227 (4QpsJub^a-c).

46 Cf. Danker et al. (2000, pp. 853, 1032). Gal 1:13b (RSV) translates "how I persecuted the church of God *violently* and tried to destroy it".

47 It may not be a coincidence that φόνος, killing, something "breathed as a threat" by Saul in Acts 9:1, also parallels the readiness for bloodshed on the part of the radicalized milieu of the "Fourth Philosophy", as described by Josephus, *Ant.* 18.5, 18.8, 18.23.

48 See, e.g., the survey by Elmer (2009, pp. 131–62) at 132 on the difficulty to "'mirror-read' Paul's comments in order to reconstruct both the identity of his opponents and the content of their gospel".

49 Dunn (1993, p. 97) compares Paul's treatment of "false brothers" with how he treated "the other gospel" (Gal 1:6–9); Keener 2019, pp. 118–19, interprets Gal 2:4 as a Pauline metaphor "used figuratively for rhetorical strategy".

[50]   Keener (2019, p. 388) compares Gal 4:17 with the motif of the "excluded lover". Yet, in a sectarian context, the act of excluding people from fellowship could be a severe measure of imposing communal boundary markings (cf., e.g., 1QS 5.10–13; Josephus, *J.W.* 2.143–144 on Essene communal boundary marking).

[51]   Cf. Fitzmyer (1998, pp. 129–52) on the Lucan story of Paul, at pp. 145–47, about the "Paulinism of Acts", including discussion of scholarly positions on differences between the Lucan Paul of Acts and Paul in his own words in his Letters, critiquing the view of P. Vielhauer about these differences for being "clearly exaggerated" (p. 147).

[52]   Elsewhere, in *Ant.* 20.186, Josephus explains the name σικάριοι as being derived from their use of small daggers, called sicae by the Romans, to attack people in Judaea.

[53]   Josephus, *J.W.* 2.253, also narrates that, even before the rise of the Sicarii in the 50s CE, Judaea had been festered by brigands led by one Eleazar for a period of twenty years, a problem dealt with by Festus, once he had been installed as procurator (*J.W.* 2.252), by arresting them and transporting them to Rome for trial.

[54]   In this passage in Josephus, cf. McLaren (2001, pp. 1–25), who argues that James fell victim to rival priestly factions in Jerusalem.

[55]   As for the radicalized milieu of Sicarii, Josephus, *J.W.* 7.254–255, observes that their ultimate designs were to treat fellow Jews who did not fight for freedom but were subjected as slaves to Rome as no different from foreigners, devastating them and their environment.

[56]   Cf. 4Q524 (*4QTemple Scroll*) frg. 14 ll. 2–4//11QTᵃ col. 64 ll. 6–13, in particular line 12, for the idea of a crucified person as "cursed by God and man".

[57]   Cf. Rom 3:8 (RSV), "And why not do evil that good may come?—as some people slanderously charge us with saying. Their condemnation is just". Rom 3:31 (RSV), "Do we then overthrow the law by this faith? By no means! On the contrary, we uphold the law".

[58]   Texts and translations from García Martínez and Tigchelaar (2000, pp. 952–53, 1080–81, 1120–21).

[59]   4QpNah 3–4 ii 4–6 further envisions an ominous fate for "the rule of those looking for easy interpretations" being oppressed by "the sword of the gentiles", captivity, looting, exile, massive bloodshed, "because of their guilty counsel". Translation from García Martínez and Tigchelaar (2000, p. 339).

[60]   Cf. Hebrew blessings of אל ישראל in, e.g., 4Q502 frgs. 7–10 ll. 5, 10, 16, frg. 14 l. 4, frg. 24 l. 2; 4Q503 (*4QDaily Prayersᵃ*) *passim*.

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
