# Peer review of "“Our Freedom in Christ”: Revisiting Pauline Imagery of Freedom and Slavery in His Letter to the Galatians in Context"

_religions, doi:10.3390/rel14050672_

Round 1

Author Response

Dear reviewer,

Thank you for your three valuable suggestions for typographical improvement, which I have incorporated in my manuscript.

Kind regards,

The author

Reviewer 2 Report

This is a wonderfully contextualized study of freedom as presented in Galatians, with what for me were new angles of vision from which to view some aspects of Paul's possible thought processes. 

My major concern, which can be modified easily enough, appears especially toward the conclusion, including the conclusion itself: the use of extra-Galatians texts to furbish the main point that Paul is urging a freedom from slavery to "intrusive influecnes [sic]" due to "the encroachment of a radical milieu of thought in Jerusalem." The author cites the Paul and his issues as presented by Luke, in Acts and finishes in his final paragraph with the Paul of the first letter to Corinth. Luke's text of Luke-Acts presents in Acts quite a different Paul and Paul's thinking, and even circumstances, than we find in any of his writings, including most certainly the Paul of Galatians.

I'm not sure to what extent the author's conclusions depend on these extra-textual sources, but I think their use weakens his argument and would advice amending the article by explicitly qualifying the use of these other sources or eliminating them entirely. With such amendment I would recommend the article's publication.

Author Response

Dear reviewer,

Thank you for your detailed feedback and recommendation for the article's publication after amendment of the point you made. I have made the amendment by inserting the following small paragraph in the text at the point where you uttered your critical concern:

"What Paul’s Letter to the Galatians lacks in further explicit detail regarding Jerusalemite circles apart from hints at compulsion, external pressure and persecution may be compared with external contextual evidence of Flavius Josephus on the Fourth Philosophy, as we have seen above (§ 5.1 above). The fact that milieus of radicalization in Jerusalem would eventually touch the context of Paul’s ultimate visit of Jerusalem may be gathered from the secondary evidence of Luke’s story of Paul in Acts 21:17-23:22. Even though the evidence of Acts may by no means constitute primary evidence of Paul’s life and thought, the author of Acts did have his sources of information about Paul beyond the timeframe of his letters.[i] His narrative does provide an additional perspective on the subject of radicalized milieus in Jerusalem bythe late 50s CE."

[i]         Cf. Fitzmyer 1998, pp. 129-52 on the Lucan story of Paul, at pp. 145-47 about the ‘Paulinism of Acts’, including discussion of scholarly positions on differences between the Lucan Paul of Acts and Paul in his own words in his Letters, critiquing the view of P. Vielhauer about these differences for being “clearly exaggerated” (p. 147).

After this paragraph, I have continued the text as I already had it. I hope that this amendment meets your concern of "explicitly qualifying the use of these other sources [Acts]".

Kind regards,

The author

Reviewer 3 Report

This is a well-argued and well-supported discussion on Paul's use of freedom language throughout Galatians. Overall, the article is coherent, consistent and clear, which made for a highly enjoyable read.  The author's engagement with and appropriate use of primary texts is excellent and, I argue, correctly and accurately used throughout the article. Their subsequent discussion on how these might support their own understanding and interpretation of Paul's use of similar ideas, phrases and words in Galatians is also excellent, leading me to agree with the author's overall conclusions, even when I have come to alternate interpretations in my own work.  

Author Response

Dear reviewer,

Many thanks for your very positive feedback about my article. This further encourages me to resubmit it for publication.

Kind regards,

The author